# FINBERT: FINANCIAL SENTIMENT ANALYSIS WITH PRE-TRAINED LANGUAGE MODELS

## ABSTRACT

While many sentiment classification solutions report high accuracy scores in product or movie review datasets, the performance of the methods in niche domains such as finance still largely falls behind. The reason of this gap is the domain-specific language, which decreases the applicability of existing models, and lack of quality labeled data to learn the new context of positive and negative in the specific domain. Transfer learning has been shown to be successful in adapting to new domains without large training data sets. In this paper, we explore the effectiveness of NLP transfer learning in financial sentiment classification. We introduce FinBERT, a language model based on BERT, which improved the state-of-the-art performance by 14 percentage points for a financial sentiment classification task in FinancialPhrasebank dataset.

## 1 INTRODUCTION

With unprecedented amount of textual data being created every day, analyzing large bodies of text from distinct domains like medicine or finance is of the utmost importance. Yet it is more difficult to apply supervised NLP methods, like text classification, in these domains than it is for more general language. The difficulty comes from two factors: 1) The most sophisticated classification methods that make use of neural nets require vast amounts of labeled data and labeling domain-specific text snippets requires costly expertise. 2) The NLP models trained on general corpora are not well-suited to supervised tasks since domain-specific texts have a specialized language with unique vocabulary and expressions.

NLP transfer learning methods look like a promising solution to both of the challenges mentioned above, and are the focus of this paper. The core idea behind these models is first training a language model on a very large corpus and then initializing down-stream models with the weights learned from the language modeling task. The initialized layers can range from the single word embedding layer Peters et al. (2018) to the whole model Howard & Ruder (2018). This approach should reduce the size of the required labeled data since language models learn the language syntax and semantic in an unsupervised way on a very large unlabeled corpora by predicting the next word. By further pre-training a language model on a domain specific unlabeled corpus, the model can learn the semantic relations in the text of the target domain, which is likely to have a different distribution than a general corpus.

In this paper, we explore the effectiveness of using and fine-tuning a pre-trained language model, BERT Devlin et al. (2018), in financial sentiment classification using the Financial PhraseBank created by Malo et al. (2014) and FiQA Task-1 sentiment scoring dataset in Maia et al. (2018b).

The main contributions of this paper are the following:

- We introduce FinBERT, which is a language model based on BERT for financial NLP tasks. We evaluate FinBERT on two financial sentiment analysis datasets, where we achieve the state-of-the-art on FiQA sentiment scoring and Financial PhraseBank.

- We implement two other pre-trained language models, ULMFit and ELMo for financial sentiment analysis and compare these with FinBERT.

- We conduct experiments to investigate several aspects of the model, including: effects of further pre-training on financial corpus, training strategies to prevent catastrophic forgetting

and fine-tuning only a small subset of model layers for decreasing training time without a significant drop in performance.

## 2 FINBERT

BERT (Devlin et al., 2018) is a language model that consists of a set of transformer encoders (Vaswani et al., 2017) stacked on top of each other. It defines the language modeling in a novel way. Instead of predicting the next word given previous ones, BERT "masks" a randomly selected 15% of all tokens. With a softmax layer over vocabulary on top of the last encoder layer the masked tokens are predicted. A second task BERT is trained on is "next sentence prediction". Given two sentences, the model predicts whether or not these two actually follow each other.

Following the previous work (Howard & Ruder, 2018) on the effectiveness of further pre-training a language model on a target domain, we experimented with two approaches: The first is pre-training the model on a relatively large corpus from the target domain. For that, we further pre-train a BERT language model on a financial corpus. The second approach is pre-training the model only on the sentences from the training classification dataset. Although the second corpus is much smaller, using data from the direct target might provide better target domain adaptation.

Sentiment classification is conducted by adding a dense layer after the last hidden state of the [CLS] token. This is the recommended practice for using BERT for any classification task (Devlin et al., 2018). Then, the classifier network is trained on the labeled sentiment dataset. An overview of all the steps involved in the procedure is presented on figure 2.

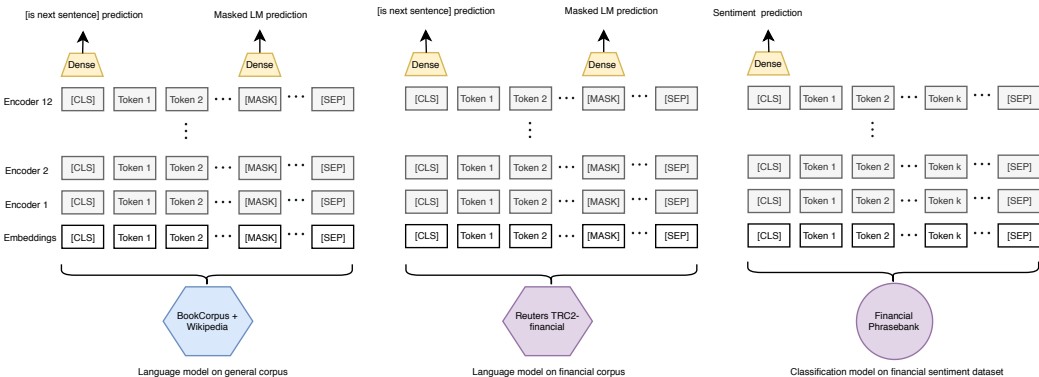

Figure 1: Sample figure caption.

While the focus of this paper is classification, we also implement regression with almost the same architecture on a different dataset with continuous targets. The only difference is that the loss function being used is mean squared error instead of the cross entropy loss.

As it was pointed out by Howard & Ruder (2018), catastrophic forgetting is a significant danger with this fine-tuning approach. Because the fine-tuning procedure can quickly cause model to "forget" the information from language modeling task as it tries to adapt to the new task. In order to deal with this phenomenon, we apply three techniques as it was proposed by Howard & Ruder (2018): slanted triangular learning rates, discriminative fine-tuning and gradual unfreezing.

Slanted triangular learning rate applies a learning rate schedule in the shape of a slanted triangular, that is, learning rate first linearly increases up to some point and after that point linearly decreases.

Discriminative fine-tuning is using lower learning rates for lower layers on the network. Assume our learning rate at layer $l$ is $\alpha$. Then for discrimination rate of $\theta$ we calculate the learning rate for layer $l-1$ as $\alpha_{l-1} = \theta\alpha_l$. The assumption behind this method is that the lower layers represent the deep-level language information, while the upper ones include information for actual classification task. Therefore we fine-tune them differently.

Table 1: Distribtution of sentiment labels and agreement levels in Financial PhraseBank

| Agreement level | Positive | Negative | Neutral | Count |
|---|---|---|---|---|
| 100% | 25.2% | 13.4% | 61.4% | 2262 |
| 75% - 99% | 26.6% | 9.8% | 63.6% | 1191 |
| 66% - 74% | 36.7% | 12.3% | 50.9% | 765 |
| 50% - 65% | 31.1% | 14.4% | 54.5% | 627 |
| All | 28.1% | 12.4% | 59.4% | 4845 |

With gradual freezing, we start training with all layers but the classifier layer as frozen. During training we gradually unfreeze all of the layers starting from the highest one, so that the lower level features become the least fine-tuned ones. Hence, during the initial stages of training it is prevented for model to "forget" low-level language information that it learned from pre-training.

## 3 EXPERIMENTAL SETUP

### 3.1 DATASETS

In order to further pre-train BERT, we use a financial corpus we call TRC2-financial. It is a subset of Reuters' TRC2[1], which consists of 1.8M news articles that were published by Reuters between 2008 and 2010. We filter for some financial keywords in order to make corpus more relevant and in limits with the compute power available. The resulting corpus, TRC2-financial, includes 46,143 documents with more than 29M words and nearly 400K sentences.

The main sentiment analysis dataset used in this paper is Financial PhraseBank[2] from Malo et al. 2014 Malo et al. (2014). Financial Phrasebank consists of 4845 english sentences selected randomly from financial news found on LexisNexis database. These sentences then were annotated by 16 people with background in finance and business. The annotators were asked to give labels according to how they think the information in the sentence might affect the mentioned company stock price. The dataset also includes information regarding the agreement levels on sentences among annotators. The distribution of agreement levels and sentiment labels can be seen on table 1. We set aside 20% of all sentences as test and 20% of the remaining as validation set. In the end, our train set includes 3101 examples. For some of the experiments, we also make use of 10-fold cross validation.

FiQA Maia et al. (2018b) is a dataset that was created for WWW '18 conference financial opinion mining and question answering challenge[3]. We use the data for Task 1, which includes 1,174 financial news headlines and tweets with their corresponding sentiment score. Unlike Financial Phrasebank, the targets for this datasets are continuous ranging between $[-1, 1]$ with 1 being the most positive. Each example also has information regarding which financial entity is targeted in the sentence. We do 10-fold cross validation for evaluation of the model for this dataset.

### 3.2 BASELINE METHODS

For contrastive experiments, we consider baselines with three different methods: LSTM classifier with GLoVe embeddings, LSTM classifier with ELMo embeddings and ULMFit classifier. It should be noted that these baseline methods are not experimented with as thoroughly as we did with BERT. Therefore the results should not be interpreted as definitive conclusions of one method being better.

We implement two classifiers using bidirectional LSTM models. In both of them, a hidden size of 128 is used, with the last hidden state size being 256 due to bidirectionality. A fully connected feed-forward layer maps the last hidden state to a vector of three, representing likelihood of three

---

[1]The corpus can be obtained for research purposes by applying here: https://trec.nist.gov/data/reuters/reuters.html

[2]The dataset can be found here: https://www.researchgate.net/publication/251231364 _FinancialPhraseBank-v10

[3]Data can be found here: https://sites.google.com/view/fiqa/home

Table 2: Experimental Results on the Financial PhraseBank dataset

| Model | All data | | | Data with 100% agreement | | |
|---|---|---|---|---|---|---|
| | Loss | Accuracy | F1 Score | Loss | Accuracy | F1 Score |
| LSTM | 0.81 | 0.71 | 0.64 | 0.57 | 0.81 | 0.74 |
| LSTM with ELMo | 0.72 | 0.75 | 0.7 | 0.50 | 0.84 | 0.77 |
| ULMFit | 0.41 | 0.83 | 0.79 | 0.20 | 0.93 | 0.91 |
| LPS | - | 0.71 | 0.71 | - | 0.79 | 0.80 |
| HSC | - | 0.71 | 0.76 | - | 0.83 | 0.86 |
| FinSSLX | - | - | - | - | 0.91 | 0.88 |
| FinBERT | **0.37** | **0.86** | **0.84** | **0.13** | **0.97** | **0.95** |

LPS (Malo et al., 2014), HSC (Krishnamoorthy, 2018) and FinSSLX (Maia et al., 2018b) results are taken from their respective papers. For LPS and HSC, overall accuracy is not reported on the papers. We calculated them using recall scores reported for different classes. For the models implemented by us, we report 10-fold cross validation results.

labels. The difference between two models is that one uses GLoVe embeddings, while the other uses ELMo embeddings. A dropout probability of 0.3 and a learning rate of 3e-5 is used in both models. We train them until there is no improvement in validation loss for 10 epochs.

Classification with ULMFit consists of three steps. The first step of pre-training a language model is already done and the pre-trained weights are released by Howard and Ruder (2018). We first further pre-train AWD-LSTM language model on TRC2-financial corpus for 3 epochs. After that, we fine-tune the model for classification on Financial PhraseBank dataset, by adding a fully-connected layer to the output of pre-trained language model.

### 3.3 IMPLEMENTATION DETAILS

For our implementation BERT, we use a dropout probability of $p = 0.1$, warm-up proportion of 0.2, maximum sequence length of 64 tokens, a learning rate of $2e - 5$ and a mini-batch size of 64. We train the model for 6 epochs, evaluate on the validation set and choose the best one. For discriminative fine-tuning we set the discrimination rate as 0.85. We start training with only the classification layer unfrozen, after each third of a training epoch we unfreeze the next layer. An Amazon p2.xlarge EC2 instance with one NVIDIA K80 GPU, 4 vCPUs and 64 GiB of host memory is used to train the models.

## 4 EXPERIMENTAL RESULTS

The results of FinBERT, the baseline methods and state-of-the-art on Financial PhraseBank dataset classification task can be seen on table 2. We present the result on both the whole dataset and subset with 100% annotator agreement.

For all of the measured metrics, FinBERT performs clearly the best among both the methods we implemented ourselves (LSTM and ULMFit) and the models reported by other papers (LPS Malo et al. (2014), HSC Krishnamoorthy (2018), FinSSLX Maia et al. (2018a)). LSTM classifier with no language model information performs the worst. In terms of accuracy, it is close to LPS and HSC, (even better than LPS for examples with full agreement), however it produces a low F1-score. That is due to it performing much better in neutral class. LSTM classifier with ELMo embeddings improves upon LSTM with static embeddings in all of the measured metrics. It still suffers from low average F1-score due to poor performance in less represented labels. But it's performance is comparable with LPS and HSC, besting them in accuracy. So contextualized word embeddings produce close performance to machine learning based methods for dataset of this size.

ULMFit significantly improves on all of the metrics and it doesn't suffer from model performing much better in some classes than the others. It also handily beats the machine learning based models LPS and HSC. This shows the effectiveness of language model pre-training. AWD-LSTM is a very

Table 3: Experimental Results on FiQA Sentiment Dataset

| Model | MSE | $R^2$ |
|---|---|---|
| Yang et. al. (2018) | 0.08 | 0.40 |
| Piao and Breslin (2018) | 0.09 | 0.41 |
| FinBERT | **0.07** | **0.55** |

Yang et al. (2018) and Piao & Breslin (2018) report results on the official test set. Since we don't have access to that set our MSE, and $R^2$ are calculated with 10-Fold cross validation.

large model and it would be expected to suffer from over-fitting with this small of a dataset. But due to language model pre-training and effective training strategies, it is able to overcome small data problem. ULMFit also outperforms FinSSLX, which has a text simplification step as well as pre-training of word embeddings on a large financial corpus with sentiment labels.

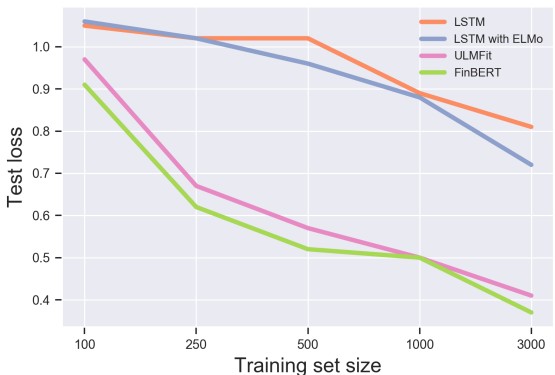

Figure 2: Test loss different training set sizes

FinBERT outperforms ULMFit, and consequently all of the other methods in all metrics. In order to measure the performance of the models on different sizes of labeled training datasets, we ran LSTM classifiers, ULMFit and FinBERT on 5 different configurations. The result can be seen on figure 2, where the cross entropy losses on test set for each model are drawn. 100 training examples is too low for all of the models. However, once the training size becomes 250, ULMFit and FinBERT starts to successfully differentiate between labels, with an accuracy as high as 80% for FinBERT. All of the methods consistently get better with more data, but ULMFit and FinBERT does better with 250 examples than LSTM classifiers do with the whole dataset. This shows the effectiveness of language model pre-training.

The results for FiQA sentiment dataset, are presented on table 3. Our model outperforms state-of-the-art models for both MSE and $R^2$. It should be noted that the test set these two papers Yang et al. (2018) Piao & Breslin (2018) use is the official FiQA Task 1 test set. Since we don't have access to that we report the results on 10-Fold cross validation. There is no indication on Maia et al. (2018b) that the train and test sets they publish come from different distributions and our model can be interpreted to be at disadvantage since we need to set aside a subset of training set as test set, while state-of-the-art papers can use the complete training set.

## 5 EXPERIMENTAL ANALYSIS

### 5.1 EFFECTS OF FURTHER PRE-TRAINING

We first measure the effect of further pre-training on the performance of the classifier. We compare three models: 1) No further pre-training (denoted by Vanilla BERT), 2) Further pre-training on

Table 4: Performance with different pre-training strategies

| Model | Loss | Accuracy | F1 Score |
|---|---|---|---|
| Vanilla BERT | 0.38 | 0.85 | 0.84 |
| FinBERT-task | 0.39 | 0.86 | **0.85** |
| FinBERT-domain | **0.37** | **0.86** | 0.84 |

Results are reported on 10-fold cross validation.

classification training set (denoted by FinBERT-task), 3) Further pre-training on domain corpus, TRC2-financial (denoted by FinBERT-domain). Models are evaluated with loss, accuracy and macro average F1 scores on the test dataset. The results can be seen on table 4.

The classifier that were further pre-trained on financial domain corpus performs best among the three, though the difference is not very high. There might be four reasons behind this result: 1) The corpus might have a different distribution than the task set, 2) BERT classifiers might not improve significantly with further pre-training, 3) Short sentence classification might not benefit significantly from further pre-training, 4) Performance is already so good, that there is not much room for improvement. We think that the last explanation is the likeliest, because for the subset of Financial Phrasebank that all of the annotators agree on the result, accuracy of Vanilla BERT is already 0.96. The performance on the other agreement levels should be lower, as even the humans can't agree fully on them. More experiments with another financial labeled dataset is necessary to conclude that effect of further pre-training on domain corpus is not significant.

## 5.2 CATASTROPHIC FORGETTING

For measuring the performance of the techniques against catastrophic forgetting, we try four different settings: No adjustment (NA), only with slanted triangular learning rate (STL), slanted triangular learning rate and gradual unfreezing (STL+GU) and the techniques in the previous one, together with discriminative fine-tuning. We report the performance of these four settings with loss on test function and trajectory of validation loss over training epochs. The results can be seen on table 5 and figure 3.

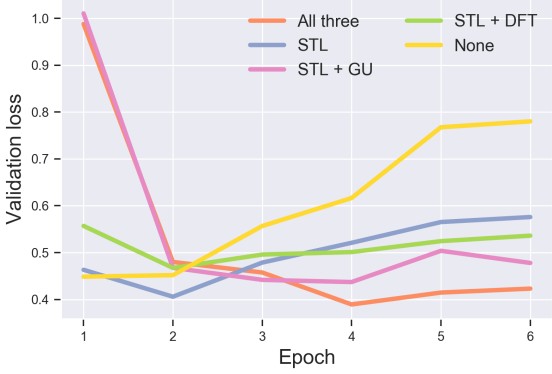

Figure 3: Validation loss trajectories with different training strategies

Applying all three of the strategies produce the best performance in terms of test loss and accuracy. Gradual unfreezing and discriminative fine-tuning have the same reasoning behind them: higher level features should be fine-tuned more than the lower level ones, since information learned from language modeling are mostly present in the lower levels. We see from table 5 that using only discriminative fine-tuning with slanted triangular learning rates performs worse than using the slanted triangular learning rates alone. This shows that gradual unfreezing is the most important technique for our case.

Table 5: Performance with different fine-tuning strategies

| Strategy | Loss | Accuracy | F1 Score |
|----------|------|----------|----------|
| None | 0.48 | 0.83 | 0.83 |
| STL | 0.40 | 0.81 | 0.82 |
| STL + GU | 0.40 | 0.86 | **0.86** |
| STL + DFT | 0.42 | 0.79 | 0.79 |
| All three | **0.37** | **0.86** | 0.84 |

Results are reported on 10-fold cross validation. STL: slanted triangular learning rates, GU: gradual unfreezing, DFT: discriminative fine-tuning.

Table 6: Performance on starting training from different layers

| First layer unfreezed | Loss | Accuracy | Training time |
|-----------------------|------|----------|---------------|
| Embeddings layer | 0.37 | 0.86 | 332s |
| Layer-1 | 0.39 | 0.83 | 302s |
| Layer-2 | 0.39 | 0.83 | 291s |
| Layer-3 | 0.38 | 0.83 | 272s |
| Layer-4 | 0.38 | 0.82 | 250s |
| Layer-5 | 0.40 | 0.83 | 240s |
| Layer-6 | 0.40 | 0.81 | 220s |
| Layer-7 | 0.39 | 0.82 | 205s |
| Layer-8 | 0.39 | 0.84 | 188s |
| Layer-9 | 0.39 | 0.84 | 172s |
| Layer-10 | 0.41 | 0.84 | 158s |
| Layer-11 | 0.45 | 0.82 | 144s |
| Layer-12 | 0.47 | 0.81 | 133s |
| Classification layer | 1.04 | 0.52 | 119s |

One way that catastrophic forgetting can show itself is the sudden increase in validation loss after several epochs. As model is trained, it quickly starts to overfit when no measure is taken accordingly. As it can be seen on the figure 3, that is the case when none of the aforementioned techniques are applied. The model achieves the best performance on validation set after the first epoch and then starts to overfit. While with all three techniques applied, model is much more stable. The other combinations lie between these two cases.

### 5.3 TRAINING ONLY A SUBSET OF THE LAYERS

BERT is a very large model. Even on small datasets, fine-tuning the whole model requires significant time and computing power. Therefore if a slightly lower performance can be achieved with fine-tuning only a subset of all parameters, it might be preferable in some contexts. Especially if training set is very large, this change might make BERT more convenient to use. Here we experiment with fine-tuning only the last $k$ many encoder layers.

The results are presented on table 6. Fine-tuning only the classification layer does not achieve close performance to fine-tuning other layers. However fine-tuning only the last layer handily outperforms the state-of-the-art machine learning methods like HSC. After Layer-9, the performance becomes virtually the same, only to be outperformed by fine-tuning the whole model. This result shows that in order to utilize BERT, an expensive training of the whole model is not mandatory. A fair trade-off can be made for much less training time with a small decrease in model performance.

### 5.4 WHERE DOES THE MODEL FAIL?

With 97% accuracy on the subset of Financial PhraseBank with 100% annotator agreement, we think it might be an interesting exercise to examine cases where the model failed to predict the true

label. Therefore in this section we will present several examples where model makes the wrong prediction. Also in Malo et al. (2014), it is indicated that most of the inter-annotator disagreements are between positive and neutral labels (agreement for separating positive-negative, negative-neutral and positive-neutral are 98.7%, 94.2% and 75.2% respectively). Authors attribute that the difficulty of distinguishing "commonly used company glitter and actual positive statements". We will present the confusion matrix in order to observe whether this is the case for FinBERT as well.

**Example 1:** `Pre-tax loss totaled euro 0.3 million , compared to a loss of euro 2.2 million in the first quarter of 2005 .`

**True value:** Positive **Predicted:** Negative

**Example 2:** `This implementation is very important to the operator , since it is about to launch its Fixed to Mobile convergence service in Brazil`

**True value:** Neutral **Predicted:** Positive

**Example 3:** `The situation of coated magazine printing paper will continue to be weak .`

**True value:** Negative **Predicted:** Neutral

The first example is actually the most common type of failure. The model fails to do the math in which figure is higher, and in the absence of words indicative of direction like "increased", might make the prediction of neutral. However, there are many similar cases where it does make the true prediction too. Examples 2 and 3 are different versions of the same type of failure. The model fails to distinguish a neutral statement about a given situation from a statement that indicated polarity about the company. In the third example, information about the company's business would probably help.

73% of the failures happen between labels positive and negative, while same number is 5% for negative and positive. That is consistent with both the inter-annotator agreement numbers and common sense. It is easier to differentiate between positive and negative. But it might be more challenging to decide whether a statement indicates a positive outlook or merely an objective observation.

## 6 CONCLUSION AND FUTURE WORK

In this paper, we implemented BERT for the financial domain by further pre-training it on a financial corpus and fine-tuning it for sentiment analysis (FinBERT). This work is the first application of BERT for finance to the best of our knowledge and one of the few that experimented with further pre-training on a domain-specific corpus. On both of the datasets we used, we achieved state-of-the-art results by a significant margin. For the classification task, we increased the state-of-the art by 15% in accuracy.

In addition to BERT, we also implemented other pre-training language models like ELMo and ULM-Fit for comparison purposes. ULMFit, further pre-trained on a financial corpus, beat the previous state-of-the art for the classification task, only to a smaller degree than BERT. These results show the effectiveness of pre-trained language models for a down-stream task such as sentiment analysis especially with a small labeled dataset. The complete dataset included more than 3000 examples, but FinBERT was able to surpass the previous state-of-the art even with a training set as small as 500 examples. This is an important result, since deep learning techniques for NLP have been traditionally labeled as too "data-hungry", which is apparently no longer the case.

We conducted extensive experiments with BERT, investigating the effects of further pre-training and several training strategies. We couldn't conclude that further pre-training on a domain-specific corpus was significantly better than not doing so for our case. Our speculation is that BERT already performs good enough with our dataset that there is not much room for improvement that further pre-training can provide. We also found that learning rate regimes that fine-tune the higher layers more

aggressively than the lower ones perform better and are more effective in preventing catastrophic forgetting. Another conclusion from our experiments was that, comparable performance can be achieved with much less training time by fine-tuning only the last 2 layers of BERT.

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
