# OpenReview forum: "FINBERT:  FINANCIAL SENTIMENT ANALYSIS   WITH PRE-TRAINED LANGUAGE MODELS"
_ICLR.cc/2020/Conference — Reject_

### Official Review · AnonReviewer2 · 2019-10-22
**Official Blind Review #2**

**Rating:** 3

**Review:**

This paper proposes a domain adaptation type of task via proposing fine-tuning of pre-trained models such as BERT on data from financial domains. The paper starts off with a good motivation about requiring some kind of domain adaptation particularly when performing tasks such as sentiment analysis on data sets from the financial domain. However, there is not much novelty in this paper.

1)The authors do not propose any new model architectures. Even if we were to argue the novelty is in terms of their empirical work, there are some flaws/missing details in the experiments.
2)In table 1 authors present agreement amongst annotators, it would be nice if in addition to mentioning the source of the data, the authors included what metric was used to attain agreement. I had to read the original paper releasing the data set to figure this out.
3)Table 4 presents results that do not seem significant. It is hard to conclude if a certain pre-training strategy worked for sure.

On the whole I am very lukewarm on this paper. I find this paper lacking in novelty. Seems like an ambitious class project turned into an ICLR submission.

**Experience Assessment:**

I have published in this field for several years.

**Review Assessment: Checking Correctness Of Derivations And Theory:**

N/A

**Review Assessment: Checking Correctness Of Experiments:**

I assessed the sensibility of the experiments.

**Review Assessment: Thoroughness In Paper Reading:**

I read the paper at least twice and used my best judgement in assessing the paper.

---

### Official Review · AnonReviewer3 · 2019-10-24
**Official Blind Review #2529**

**Rating:** 3

**Review:**

This paper presents an analysis of the BERT language model on financial text. FinBERT is evaluated on two datasets from the financial domain: a sentiment prediction dataset (classification with 3 different classes) and a sentiment score prediction (the score is a float number between -1 and 1).

I find the phrasing "FinBERT is a language model based on BERT" misleading; I think FinBERT is BERT trained on financial text. There is no modification that is done to the original BERT model.

The paper presents several experiments using BERT as the language model and fine-tuning for the financial tasks. FinBERT is compared to a few baselines such as LSTMs with ElMO embeddings and ULMfit. I find interesting that the model performs better on the subset of the dataset for which there is perfect agreement between the annotators.

I also find the results on training on financial data interesting. The results seem to indicate that further training on financial text does not seem to result in additional improvement when compared to original BERT.

While I find the analysis and the experiments presented in the paper interesting, the novelty of the paper is rather low. There is no new idea introduced in this paper, it contains a series of experiments with BERT on financial text and tasks.



**Experience Assessment:**

I have published one or two papers in this area.

**Review Assessment: Checking Correctness Of Derivations And Theory:**

I assessed the sensibility of the derivations and theory.

**Review Assessment: Checking Correctness Of Experiments:**

I carefully checked the experiments.

**Review Assessment: Thoroughness In Paper Reading:**

I read the paper thoroughly.

---

### Official Review · AnonReviewer1 · 2019-10-28
**Official Blind Review #1**

**Rating:** 1

**Review:**

This paper presents a method for financial sentiment analysis based on the texts obtained from news. The method is based on an existing method BERT (Devlin et al. 2018).  The authors have performed thorough experimental studies of the BERT method on an existing dataset TRC2-financial, a subset of TRC2 consisting of 1.8M news articles. Although the results may be of interest to communities working in this area, there are no or little novel contributions. By reading section 2 (only one page), which describes the method used, I have the impression that the authors took the method BERT and then applied this to the TRC2-financial dataset and then reported the results and also discussed some parameter choices in the BERT method. Therefore, the only value about this paper is the experimental results. Apart from this, there are no other contributions or insights to the methods/problems. In addition, section 2 is over-brief and very unclear, and it only contains a brief summary of the BERT method. For these reasons, I think the paper should be rejected for lacking novelty and writing quality.

**Experience Assessment:**

I do not know much about this area.

**Review Assessment: Checking Correctness Of Derivations And Theory:**

N/A

**Review Assessment: Checking Correctness Of Experiments:**

I carefully checked the experiments.

**Review Assessment: Thoroughness In Paper Reading:**

I read the paper thoroughly.

---

### Official Review · AnonReviewer4 · 2019-11-03
**Official Blind Review #4**

**Rating:** 3

**Review:**

This paper described the application of BERT in the field of financial sentiment analysis. Authors find that when fine-tuned with in-domain data, BERT outperforms the state-of-the-art, demonstrating that language model pre-training can transfer knowledge learned from unsupervised large corpus to new domain with minimum effort. Experiments are conducted to explore 1) the utility of different in-domain dataset for further pre-training; 2) strategies to avoid catastrophic forgetting, and 3) effectiveness of fine-tuning a subset of the full model.

I am in favor of rejecting this paper and my reasons are as follows:

First, this paper may lack deeper innovation, although it demonstrates a good application of the BERT models in financial domain. For example, the framework of general-domain LM pretraining, to in-domain LM pretraining and finally in-domain classifier fine-tuning, as well as techniques of catastrophic forgetting were already proposed in Howard & Ruder 2018. Therefore, I think this paper may be more suitable for other (finance) application-oriented venues.

Second, the dataset used in evaluation is of small size (for example, Financial PhraseBank test set has one 1K). Thus, even though the paper is about transfer learning to domains without large data, I find it might be more convincing to draw a solid conclusion with a larger test set.

This paper is well organized and easy to follow. It may be beneficial to clarify in a few places (if space permits):
1) Some description or statistics of the data may be helpful (e.g., average sentence length or some examples);
2) Citations to Elmo and ULMFit can be made more explicit. Authors did cite Peters 2018 and Howard 2018 at the beginning of the paper, but may want to explicitly associate them with ‘Elmo’ and ‘ULMFit’ when these two terms first occur respectively;
3) For table 2, does the ‘all data’ or ‘data with 100% agreement’ include training data (80%) or just the test data (20%)?
The difference between FinBERT(-domain) and ULMFit can be explicitly contrasted in the paper. Is the former initialized with BERT while latter with ULMFit?


**Experience Assessment:**

I have published one or two papers in this area.

**Review Assessment: Checking Correctness Of Derivations And Theory:**

I carefully checked the derivations and theory.

**Review Assessment: Checking Correctness Of Experiments:**

I carefully checked the experiments.

**Review Assessment: Thoroughness In Paper Reading:**

I read the paper thoroughly.

---

### Decision · Program_Chairs · 2019-12-19

**Decision:**

Reject

**Comment:**

This paper presents FinBERT, a BERT-based model that is further trained on a financial corpus and evaluated on Financial PhraseBank and Financial QA. The authors show that FinBERT slightly outperforms baseline methods on both tasks.

The reviewers agree that the novelty is limited and this seems to be an application of BERT to financial dataset. There are many cases when it is okay to not present something entirely novel in terms of model as long as a paper still provides new insights on other things. Unfortunately, the new experiments in this paper are also not convincing. The improvements are very minor on small evaluation datasets, which makes the main contributions of the paper not enough for a venue such as ICLR.

The authors did not respond to any of the reviewers' concerns. I recommend rejecting this paper.